# Non-Contact Paper Thickness and Quality Monitoring Based on Mid-Infrared Optical Coherence Tomography and THz Time Domain Spectroscopy

**DOI:** 10.3390/s22041549

**Published:** 2022-02-17

**Authors:** Rasmus Eilkær Hansen, Thorsten Bæk, Simon Lehnskov Lange, Niels Møller Israelsen, Markku Mäntylä, Ole Bang, Christian Rosenberg Petersen

**Affiliations:** 1DTU Fotonik, Technical University of Denmark, 2800 Kgs. Lyngby, Denmark; raeha@fotonik.dtu.dk (R.E.H.); thbak@fotonik.dtu.dk (T.B.); slla@fotonik.dtu.dk (S.L.L.); nikr@fotonik.dtu.dk (N.M.I.); oban@fotonik.dtu.dk (O.B.); 2NORBLIS ApS, Virumgade 35 D, 2830 Virum, Denmark; 3Valmet Oyj, Keilasatama 5, FI-02150 Espoo, Finland; markku.mantyla@valmet.com

**Keywords:** OCT, TDS, THz, mid-infrared, NDT, non-contact, non-destructive, imaging, thickness, defect

## Abstract

In industrial paper production, online monitoring of a range of quality parameters is essential for ensuring that the performance and appearance of the final product is suitable for a given application. In this article, two optical sensing techniques are investigated for non-destructive, non-contact characterization of paper thickness, surface roughness, and production defects. The first technique is optical coherence tomography based on a mid-infrared supercontinuum laser, which can cover thicknesses from ~20–90 μm and provide information about the surface finish. Detection of subsurface voids, cuts, and oil contamination was also demonstrated. The second technique is terahertz time domain spectroscopy, which is used to measure paper thicknesses of up to 443 μm. A proof-of-concept thickness measurement in freely suspended paper was also demonstrated. These demonstrations highlight the added functionality and potential of tomographic optical sensing methods towards industrial non-contact quality monitoring.

## 1. Introduction

Paper is a ubiquitous product used in a wide variety of applications, from printing books and currency, to packaging and cleaning. It is generally made from a web of cellulose fibers that is processed into a thin sheet but can differ in composition and quality parameters, such as strength, density, and thickness depending on the requirements of the application. To ensure that these parameters comply with the needs of an application, industrial process control is needed. Many of these parameters can be measured directly using mechanical devices, such as calipers and scales. However, modern paper manufacturing involves a series of automated processing steps where in-line sensors are required to ensure both speed and quality in the production. To accommodate in-line sensing, it is a requirement that the sensing technique is both fast and ideally non-contact to avoid damaging the paper web. Several non-contact technologies have been implemented in industrial papermaking, including radioactive β-ray transmission [1,2] and X-ray absorption [1,3], which are primarily used to determine basis weight/density and ash content, respectively. For measurements of paper thickness, sensors based on magnetic reluctance or Eddy currents were for a long time considered the industry standard [1]. However, while magnetic sensors can provide kHz rate sampling with 1-μm accuracy, they require physical contact on both sides of the paper, which can cause issues with material build-up and sheet damage [1,4]. For this reason, optical sensors are usually considered for in-line sensing purposes. A common optical method for determining paper thickness is laser triangulation, where a laser beam is reflected from the paper surface at an angle, such that the position of the reflected beam on a detector array depends on the distance to the paper, which will vary with thickness. Today, such systems are commercially available, providing a sampling rate of several kHz and micron-level accuracy. However, since paper varies in opacity and surface roughness, the position of the reflected and backscattered signal will also vary and be distributed across the sensor, inhibiting accurate measurements. Depending on the paper type, fabrication tolerances, and mechanical stability, it may, therefore, be necessary to measure the paper from both sides to compensate for offset variations due to varying tension and vibrations. Alternatively, the paper can be brought into single-sided contact with a metallic reference surface, whose position can be accurately measured using a magnetic sensor. Using this principle, Graeffe and Nuyan verified that laser triangulation can provide similar results to contact-based, industry-standard paper machine analyzers (<±1-μm deviation) [4]. However, for absolute thickness measurements, the one-sided, laser-based measurement is still challenged by surface roughness. For example, Graeffe and Nuyan measured a one-side-coated paper sample to be 71 μm thick when measuring from the coated side, and just 58 μm from the un-coated side [4].

In this paper, two all-optical techniques are investigated for paper thickness measurements and quality inspection. The techniques are based on broadband mid-infrared (MIR) and terahertz (THz) light sources in order to penetrate the highly scattering web of cellulose fibers and additives. In the MIR region, optical coherence tomography (OCT) has recently shown great potential for non-contact inspection of scattering materials, such as ceramics and marine coatings [5,6]. OCT was originally developed for medical imaging and is based on the interference of light reflected and scattered from within a sample. Since it is based on low-intensity laser light, it can reach a microscopic resolution, is non-destructive, and does not require physical contact. Previous studies on OCT measurements of paper have been greatly limited by scattering due to the use of near-infrared light sources [7,8]. Here, by means of MIR OCT, we achieved a factor of >2 improvement in penetration depth compared to earlier studies, as well as demonstration of defect detection, surface quality monitoring, and calculation of the refractive index. In the THz region, time-domain spectroscopy (TDS) has previously been investigated for measuring both thickness, moisture, and density of paper [2,9,10]. However, due to bulky and sensitive instrumentation combined with complex data analysis, THz-TDS has not found widespread use in industrial process monitoring. Here, we investigated the potential of a compact and robust THz-TDS instrument combined with simple data processing for paper thickness measurements and compared the performance with a commercial laboratory instrument.

## 2. Materials and Methods

### 2.1. Mid-Infrared Optical Coherence Tomography

The MIR OCT system is depicted in Figure 1. It is based on a supercontinuum (SC) fiber-laser that emits over a continuous spectral band from 1–4.6 µm. After a 15-min warm-up period the laser is stabilized and shows less than 1% signal variation. The light is coupled into a free-space Michelson interferometer using a parabolic mirror, and subsequently filtered to cover just the long-wavelength region from 3.0–4.6 µm. This region was found to be useful for imaging a number of materials including ceramics, polymers, and coatings [5,6]. In the interferometer, the MIR light is split into a sample and reference path using a nitrocellulose membrane pellicle beam-splitter designed for the 3–5-µm wavelength range. Sample scanning was achieved using a two-axis, gold-coated mirror galvanometric scanner and a BaF_2_ plano-convex lens, while the reference path has two gold-coated plane mirrors and a BaF_2_ window for crude dispersion compensation. The two beams are then re-combined at the beam-splitter after being reflected from the sample and reference mirror, and the resulting interference signal is then forwarded to an upconversion spectrometer [11,12]. In the spectrometer, the interference signal is converted from the 4-µm MIR spectral region to the 0.8-µm NIR spectral region in order to benefit from the high sensitivity, resolution, and speed of NIR spectrometers. The upconverted interference signal is then coupled to a single-mode fiber and guided to a 4096-element array CMOS spectrometer, which allows for high-resolution parallel detection of the entire wavelength band with a digital sampling resolution of 4.21 µm and kHz line rate [13].

The system achieves a high axial resolution of ~8 µm due to the wide bandwidth of the laser and a lateral spatial resolution of ~30 µm limited by the lens’ focal length of 30 mm. The signal sensitivity and 6-dB sensitivity roll-off were measured to be ~60 dB and 1.35 mm, respectively. For more information about the system, see Refs. [6,13].

### 2.2. THz Time-Domain Spectroscopy

The THz measurements were performed using two separate systems in order to validate the results and investigate the influence of axial resolution. The first system (THz-DTU) was a compact prototype system made by DTU Fotonik, as shown in Figure 2b. The system is designed specifically for industrial applications, requiring both compactness, robustness, and low cost. It is based on the principle of THz cross-correlation spectroscopy (THz-CCS). THz-CCS was initially demonstrated by Morikawa et al. [14] and subsequently investigated further by Koch et al. under the more popularized name “quasi time-domain spectroscopy” [15,16,17]. In the THz-DTU system, an incoherent C-band amplified spontaneous emission light source is split into two interferometer arms. The length of one arm can be electronically controlled with respect to the other with a maximum length difference corresponding to 120 ps in free space propagation. One arm is connected to a photoconductive antenna (PCA) that transmits (Tx) THz light upon illumination from the incoherent light source. The other arm is connected to a PCA that receives THz light (Rx), which is activated upon illumination from the incoherent light source. By continuously changing the two path lengths in the interferometer, a cross-correlation trace is obtained in the Rx. This trace takes the form of a pulse, as shown in Figure 5. The light source has a bandwidth spanning from 1520 nm to 1570 nm, thus allowing an all-fiber system based on standard components developed for telecommunications. The fiber-based design makes the system highly robust with respect to environmental changes and vibrations and is, therefore, applicable in an industrial environment. The PCAs are InGaAs units for CW THz generation and detection, from TOPTICA PHOTONICS [18]. The system has a 1-THz bandwidth and a 20-dB dynamic range. The system achieved an axial resolution of ~150 µm (minimum depth separation that can be resolved) and a lateral spatial resolution of ~1 mm.

The second system (THz-TOP) was a state-of-the-art commercial benchtop instrument (Teraflash Smart, TOPTICA PHOTONICS) based on a femtosecond laser, which has a 3-THz bandwidth and ~50-dB dynamic range, allowing for a smaller minimum detectable thickness and higher resolution compared to THz-DTU. The instrument has an axial resolution of ~50 µm and a lateral spatial resolution of ~1 mm. The increased specs from the THz-TOP system come at the expense of a bulkier, more expensive, and less mechanically robust system. Since part of the THz-TOP interior is based on free-space optics, it is more sensitive to environmental conditions, and the volumetric size of the instrument is seven times larger than that of THz-DTU. All THz measurements were performed in a 20-degrees angled reflection setup, as shown in Figure 2c.

### 2.3. Paper Samples and Measurements

A set of paper samples with varying thickness, density, and composition was supplied by paper manufacturer Valmet. Each sample was accompanied by a reference thickness measured by Valmet using the single-sided laser triangulation method. A second reference measurement was performed after the optical measurements on six points of each sample using a digital micrometer (caliper) from Mitutoyo with a 1-μm accuracy. Thicknesses from 68–443 μm were tested, among which two were brown packaging papers and one had a high-gloss coated surface finish. A few of these samples are shown mounted on a sample holder in Figure 3a and optical microscope images at 20× magnification are shown in Figure 3c.

### 2.4. OCT Thickness Measurements

As shown in Figure 1, OCT imaging was performed by placing the paper samples on top of a highly reflective (>95%), silver-coated mirror to obtain the clearest possible reflection signal from the backside of the paper. This was necessary because of strong scattering and sensitivity to surface roughness at MIR wavelengths. Although not ideal, this method is similar to the one described by Graeffe, where a vacuum system is used in conjunction with a perforated metal plate to ensure physical contact at the point of measurement [19]. The OCT images used in the thickness measurements were averaged over 200 B-scans, which each consisted of 200 A-scans acquired using an integration time of 900 μs per A-scan. The measurements took 36 s and covered a 2.4 × 0.4 mm^2^ area. Since the paper thickness was based on the difference between two pixel values, the limited digital sampling resolution led to an inherent uncertainty of $\sqrt{2}\cdot 4.21$ µm ≈6 µm. To eliminate air gaps between the paper and mirror, the paper was taped firmly against the mirror. Although not as effective as applying a weak vacuum from below, it was found to be sufficient for establishing proof of concept. However, due to the possible presence of air under the paper, two different methods were considered for finding the position of the air–paper interface: 

Method 1: A running average of 10 A-scan lines was applied. The air–paper interface and the mirror position were then found from the maxima in the averaged A-scans, as shown in Figure 4b. The average value and standard deviation of 80–100 A-scan lines were then reported as the paper thickness. If it is guaranteed that no significant amount of air is present under the paper, this method is preferred.

Method 2: The air–paper interface was found in each A-scan line. The minimum value of the physical distance and the standard deviation of the neighboring 21 data points were reported as the paper thickness. This method is less sensitive to the presence of air, as the minimum value is more likely to be close to the paper–mirror interface. However, it is highly sensitive to paper defects, surface roughness, and data outliers. If air gaps are suspected to be present, this method is preferred. 

### 2.5. THz TDS Thickness Measurements

For the THz measurements with both the THz-DTU and THz-TOP systems, scattering and surface roughness had negligible impact on the signal. In comparison to MIR wavelengths, THz wavelengths were 10–1000 times longer and, therefore, scattered significantly less, but also focused down to a larger spot size [10]. The THz measurements were, therefore, performed as single-point measurements using the sample holder from Figure 3a,b. The weakly curved shape of the sample holder (Figure 3b) was designed to keep the paper in physical contact with the aluminum, thus mimicking industrially relevant measurement conditions using vacuum. Paper is known to adsorb ambient moisture, which could have an influence on the measurements [10]. All measurement were, therefore, performed under ambient conditions to obtain the most industry-relevant results. The paper was not oriented in any particular way with respect to the linear polarization of the incident THz light, although there was evidence that the fibers had a preferred orientation that induced birefringence in the THz light [20]. However, birefringence primarily alters the reflected amplitude, which was not considered for the thickness measurements. 

In the case of THz-TOP, a short THz pulse was focused onto the paper using a topaz lens with a 70-mm focal length. At the air–paper interface, the pulse was partly reflected and partly refracted (see schematic in Figure 2a). The refracted part of the pulse was completely reflected at the paper–metal interface and partly transmitted through the paper–air interface again. This resulted in two pulses arriving at the Rx unit at different times. The time delay between the pulses was then converted into paper thickness *T* using the equation
(1)$$T=\frac{c}{2\cdot n}\cdot \cos\left(\sin^{-1}\left(\frac{\sin\left(\theta \right)}{n}\right)\right)\cdot \Delta t$$
where $\theta =20^{o}$, n=1.5, and Δt is the time difference between the two pulses. The choice of n=1.5 was based on data from [10]. Both the angles of reflection and refraction were taken into consideration. Each complete time trace took 5 ms to obtain, and just a single measurement was used for thickness calculations.

In the case of THz-DTU, the THz signal focused on the paper sample was continuous wave (CW) with a particular shape in the time domain that was matched by the activating light source illuminating the Rx unit to ultimately retrieve two pulses similar to those obtained with the THz-TOP. The samples were measured in three positions, indicated as 1–3 in Figure 3a. At each position, the THz beam was moved across a 5-mm line collecting 10 measurements along the way, resulting in 30 measurements for each sample. In this case, each complete time trace took 6 s to obtain, and 10 averages were used. The long acquisition time was chosen to obtain full 120-ps delay traces that are more easily compared to other results from literature, and averaging was performed to better assess the measurement uncertainty. If only the most relevant part of the delay window is measured, and single measurements are used instead of averaging, the system can reach Hz-level acquisition rates.

## 3. Results

### 3.1. OCT Paper Thickness Measurements

Figure 4a shows an OCT B-scan on the edge of a 68-μm paper sample, capturing both the paper and the strong direct reflection from the bare mirror. The image is a maximum pixel value projection of 200 B-scans. From the OCT image, the optical thickness was calculated by finding the peaks of the surface and mirror signals, indicated by red, dashed lines. The physical thickness of the paper was found as the distance between the top of the paper and the position of the portion of bare mirror seen in the right-most part of the image of Figure 4a. A slight slope in the position of the mirror was treated by fitting a linear curve to the mirror position, in a separate image of only the mirror. 

The average intensity profile across the paper is shown in Figure 4b, indicating an optical distance of 89 μm, while the variation in thickness along the blue lines is shown in Figure 4c. The physical distance was then found by dividing the optical distance with the refractive index of the paper. However, since the refractive index of each paper sample was unknown at MIR wavelengths, a portion of the bare mirror was included in the OCT image as a calibration reference. Under normal circumstances, the reference distance would only need to be measured once. For this particular measurement, the refractive index was determined to be ~1.33 by dividing the optical length through the paper by the corresponding path length in air. This resulted in a measured thickness of 67.1 ± 2.7 μm using Method 1 and 55.7 ± 5.3 μm using Method 2, for the given measurement. In order to test the possible angular dependence of the measurement, another measurement was made further away from the paper edge, such that only a small part of the bare mirror was left in the image. This measurement yielded a thickness of 69.1 ± 4.4 μm using Method 1 and 58.5 ± 5.4 μm using Method 2. These two measurements were then repeated an additional two times at different lateral positions on the paper sample, and the average values of these six measurements are listed in Table 1 together with the measured thicknesses of the 72-μm, 83-μm, and 90-μm paper samples. For the 68-μm paper, the mirror signal was very strong and, therefore, easy to detect. However, as the paper thickness increased, the mirror signal rapidly decreased, limiting detection to a maximum of 90 μm. Results for the 90-μm paper are shown in Figure 4d–f. 

### 3.2. THz TDS Paper Thickness Measurements

Figure 5a shows the recorded time traces obtained using THz-DTU. The x-axis is the arrival time of the pulses at the detector and the y-axis is the measured electric field of the pulses. The signal from the air–paper interface was the first to arrive at the detector and, therefore, the first peak from the left in the trace. The signal from the paper–metal interface was the more intense subsequent peak, and the thickness was then calculated as the delay between the two peaks. Five papers with different thickness were investigated: 443 μm, 256 μm, 164 μm, 83 μm, and 68 μm. The results are shown in Figure 5a with graphs vertically offset for ease of comparison. Each graph has indications of the peak position of the two pulses, marked with crosses. For the 443-μm paper, positions 1–3 on the sample gave a mean thickness and standard deviation of 442 ± 9 μm, 452 ± 7 μm, and 433± 9 μm, respectively. The mean and standard deviation across all the measurements were 442.3 ± 9.5 μm. The measurements are summarized in Table 1. For the 443-μm paper, the two pulses were sufficiently separated such that simply marking the peak positions gave a paper thickness very close to the reference value. For all paper thicknesses, both pulses could be identified visually, but for thicknesses less than 443 μm, they started to interfere and eventually overlap. This made a simple thickness retrieval based on peak finding inaccurate. More complex data processing, such as the methods of Mousavi et al. [9], must therefore be considered for more accurate measurements of the absolute thickness. Figure 5b shows the corresponding measurements conducted with THz-TOP. Due to the larger bandwidth of THz-TOP, the pulses were much more separated and, therefore, provided a better estimation of the paper thickness below 256 μm. These measurements are also listed in Table 1.

Interestingly, the thickness of the 256-μm paper was significantly overestimated by both THz systems, indicating that it was not related to the bandwidth. To further investigate this issue, the 164-μm, 256-μm, and 443-μm samples were scanned with OCT. Since OCT could not penetrate these samples, they were scanned on the edge of the paper in order to measure the distance in air from the sample holder to the surface of the paper. The result is shown in Figure 6a–c. Considering the experimental conditions and the significant surface roughness of the 164-μm and 443-μm papers, the thicknesses measured with OCT was in good agreement with reference values. However, the 256-μm paper was found to be more than 100 μm thicker than the reference value, suggesting that the sample was not in proper physical contact with the sample holder and thus verifying the THz results. Looking at the blue THz-DTU trace for the 256-μm paper in Figure 6d, there was a bit of signal variation between the air–paper interface (1) and the air–metal interface (3), but the resolution of the system was not sufficient to clearly separate the peaks. Even in the red THz-TOP trace the variation was too weak to provide conclusive evidence. Nonetheless, this again suggested the presence of multiple interfaces, which could only be an air gap. The transition from paper to air was expected to give rise to an initial peak followed by a dip (opposite of the air–paper signal). This is seen from the THz-DTU trace of a suspended 443-μm paper sample in Figure 6e. While there was no interference from the sample holder signal in this trace, it was still influenced by interference between the two air–paper interface signals.

### 3.3. Surface Quality and Defect Detection

As mentioned in the introduction, surface roughness can be an issue for optical measurements of paper thickness. Here, OCT has the benefit that it can also be used to characterize the surface finish. Figure 7 shows three single B-scans acquired in 180 ms each, illustrating the significantly different surface finish of the paper samples. The white, 100-μm-thick paper in Figure 7a is clearly much more uniform compared to the brown, 443-μm paper in Figure 7b, which exhibited tens of microns in variation. A variation of this magnitude will likely result in a large variation in measured thickness when using traditional optical sensors. Figure 7c shows the 256-μm paper sample, which appears to be coated on the surface. The coating was apparent from the smooth, high-reflectivity profile of the surface combined with a thin, dark line just below. From the gap between the coating and the paper, the layer was estimated to have an optical thickness of about 30–40 μm; assuming that the coating had a refractive index of ~1.7 at 4 μm similar to that of high-gloss alkyd enamel [5], the physical thickness was on the order of 18–24 μm. This is consistent with the coating thickness of commercial photo paper reported by Prykari et al. [7]. 

While surface examination is useful for process monitoring, the real strength of OCT lies in its ability to also characterize the subsurface environment. Figure 8a shows detection of an air-gap/void in the 100-μm paper sample. From the corresponding volume scan, in Figure 8c, additional voids of varying sizes were identified. Another example of a possible paper defect is shown in Figure 8b, where a small cut was made into a paper sample and subsequently imaged with OCT. Figure 8d shows the shape and size of the cut below the surface, which could, for example, be used to identify its origin in the process. 

The final test case was an oil contamination, which is shown in Figure 9. To test the influence of oil contamination on the OCT images, a drop of sunflower oil was placed on a 68-μm paper sample using a pipette. The oil droplet is clearly seen in both the B-scans of Figure 9a,b and in the surface projection of Figure 9c. It was also evident that the oil was absorbed into the paper in a region surrounding the droplet, since the presence of oil changed both the surface reflections, seen in Figure 9c, but also provided a much stronger reflection from the mirror, as seen in the subsurface projection in Figure 9d. This was the case because the oil entered the porous structure of the paper, filling the air gaps between the fibers and thereby reducing the scattering. It effectively acted as an index-matching clearing agent. 

## 4. Discussion

It is clear from this study that optical characterization of paper based on a combination of transmission, scattering, and reflection poses different challenges for the different wavelength regimes. In the mid-infrared, OCT is effectively limited by scattering and surface roughness. In a 2005 study, E. Alarousu et al. also investigated OCT in paper measurements using a ~830-nm superluminescent diode with a 20-nm bandwidth. In their work, the authors were able to image through ~102-μm-thick copying paper by using benzyl alcohol as an optical clearing agent [8]. The alcohol acted similarly to the oil in Figure 9, in the sense that it filled the porous structure of the paper and reduced scattering. While this is interesting as a laboratory method for investigating paper samples, optical clearing is not compatible with industrial processes. In 2010, the same group used a near-infrared SC laser to achieve submicron resolution for improved surface characterization and also managed to image through uncleared pergamin paper with a thickness of 42 μm [7]. In this work, the much longer wavelength of the MIR SC laser enabled OCT imaging of standard paper samples with up to 90-μm thickness. In fact, compact SC lasers emitting up to around 10 μm have been demonstrated [21,22]. Using such long wavelengths in the mid-infrared is expected to further reduce scattering and, thus, increase the imaging depth. Furthermore, for coated papers and papers with a higher density such as pergamin, OCT is expected to perform better as the scattering from surface roughness and porous interfaces is reduced [8]. Considering the strong scattering and limited control over the physical contact between paper and mirror, the OCT thickness measurements came very close to the reference and caliper measurements for the 68-μm sample using analysis Method 1 (69.6 ± 3.2 μm) and the remaining samples using Method 2. The 72-μm paper was only slightly overestimated by Method 1 (77.4 ± 2.8 μm) and slightly underestimated by Method 2 (70.5 ± 3.7 μm), which suggested little influence of air compared to the measurement uncertainty. On the other hand, for the 83-μm and 90-μm samples, Method 1 severely overestimated the thicknesses (117.3 ± 3.3 μm and 105.7 ± 4.3 μm, respectively), while Method 2 came close and within the measurement uncertainty to the reference values (80.0 ± 7.4 μm and 97.9 ± 6.1 μm, respectively). This, coupled with the higher standard deviation in the measurements, suggested a significant influence from poor contact with the mirror. Another indicator of poor contact was the calculated refractive index for the different samples. Using Method 1, the refractive index of the 68-μm and 72-μm samples was estimated to be *n* = 1.28 and *n* = 1.32, respectively, while for the 83-μm and 90-μm samples, it was estimated to be *n* = 1.51 and *n* = 1.53, respectively. Using Method 2, the refractive indices were reduced to *n* = 1.33 and *n* = 1.42, respectively, which were both closer to the 68-μm and 72-μm samples and more reasonable values considering that the refractive index of pure cellulose is around 1.55 in the near-infrared [8].

In the THz regime, scattering is not a concern, but rather the issue is with accurately interpreting the time-domain signal and extracting the thickness information. Even with clearly defined peaks, both systems overestimated the thickness of the thinner samples. By comparing the results in Table 1, it was evident that the high-resolution THz-TOP system was significantly closer to the reference values in the thin paper samples. Similar data processing schemes as those suggested for THz-DTU can also be used in this case to potentially yield better results, especially for the thinner paper samples. In both THz-DTU and THz-TOP, the thicknesses were overestimated, which was, in part, attributed to the presence of a small gap of air between the paper and the metal. Another factor was the uncertainty in the refractive index, which was assumed to be *n* = 1.5 for all samples across the entire bandwidth. In Mousavi et al., the measured refractive index of standard photocopy paper kept in ambient conditions (5% moisture) changed from *n* = 1.41 to *n* = 1.44 (±0.02) after oven drying, and in a finer grade of paper it changed from *n* = 1.47 to *n* = 1.50 [9]. To put it into perspective, a refractive index difference of 0.05 results in around a 14-μm difference in calculated thickness for the 443 μm using Equation (1). Improved accuracy may also be achieved by employing more advanced data processing based on numerical modeling and fitting algorithms [9]. However, the aim of this study was to investigate the accuracy of THz-TDS using simple peak-finding.

A weakness for both OCT and THz TDS is that absolute measurements of paper thickness require knowledge of the refractive index, which requires calibration. However, in many cases, the absolute thickness is less important compared to monitoring variations in the thickness [4]; for this purpose, the self-referenced nature of OCT and THz TDS could potentially be useful for process monitoring. For accurate thickness measurements alone, it is unlikely that OCT or THz TDS will ever beat laser triangulation, so the real strength lies in the ability to sense under the surface. Both techniques are applicable for on-line process control and are complementary in the range of thicknesses they can cover and in the level of details that can be resolved [23]. In thick papers, OCT can complement THz thickness measurements by adding information about surface roughness, coating/ink penetration, and defects. For very thin papers, THz also can provide spectroscopic information about moisture content and density [9,10]. Therefore, while MIR OCT could find applications within laboratory testing and development of specialty paper, for example, ink penetration, the kHz acquisition rate also allows for dense sampling of paper moving at speeds of meters per second, making it an industrially relevant technology. The THz systems are currently limited to Hz-rate sampling; on the other hand, they offer the possibility of true non-contact measurements of thickness and other quality parameters such as moisture and grammage [2]. Such a system could be valuable in measurements of, for example, crude, recycled packaging papers, where any physical contact can cause a build-up of material, which results in incorrect measurements and potential damage to the paper.

## Figures and Tables

**Figure 1 sensors-22-01549-f001:**
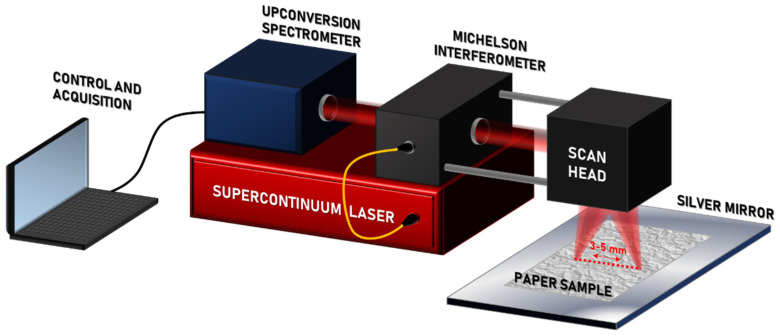
Diagram of the MIR OCT measurement setup.

**Figure 2 sensors-22-01549-f002:**
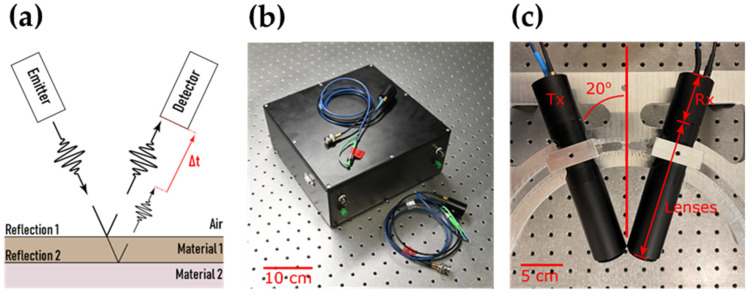
THz TDS principle and setup. (**a**) Outline of the time-of-flight principle where the delay (Δt) between reflections from the top and bottom of Material 1 (e.g., paper) is used to determine the thickness of the material. (**b**) Photograph of the complete THz-DTU measurement system, courtesy of Bax Lindhardt. (**c**) THz emitter (Tx) and detector (Rx) setup with indication of measurement angle.

**Figure 3 sensors-22-01549-f003:**
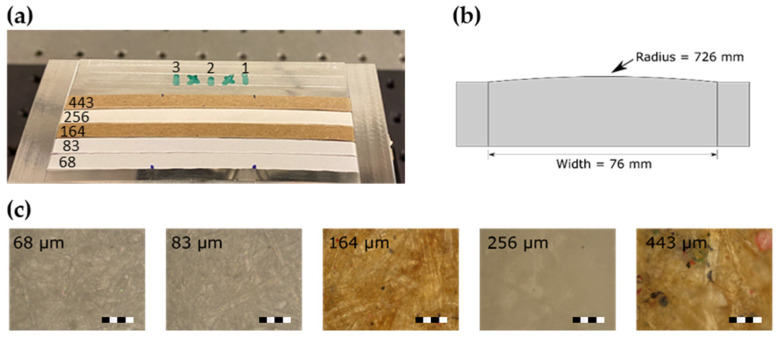
Paper samples. (**a**) Photograph of the sample holder with paper samples attached. The numbers 1, 2, and 3 indicate the lines along which measurements were taken and averaged. (**b**) Sketch of the arched sample holder, designed to stretch the paper and allow for better physical contact between paper and metal. (**c**) Optical microscope images of five paper samples at 20× magnification. Scale bars are 100 µm long.

**Figure 4 sensors-22-01549-f004:**
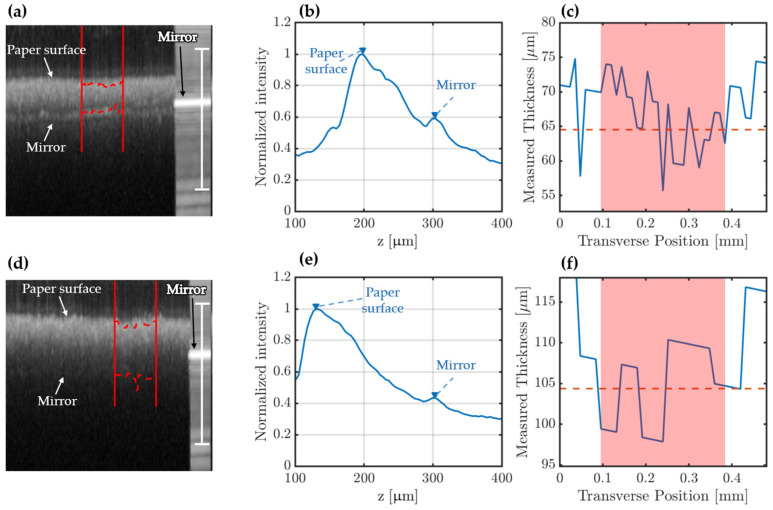
OCT measurements of paper thickness. (**a**) Average of 200 OCT B-scans on the edge of a 68-μm paper sample, capturing the strong direct reflection from the mirror. The vertical white scale bar is 0.5 mm in air. (**b**) Average transverse intensity profile from 15 A-scans. (**c**) Variation in measured paper optical thickness across the blue dashed line of (**a**). (**d**–**f**) Same as (**a**–**c**) for the 90-μm paper sample.

**Figure 5 sensors-22-01549-f005:**
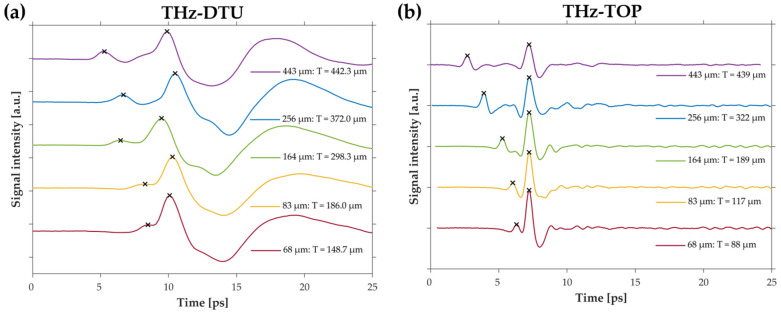
THz-TDS measurements of paper samples with 68–443-μm thickness. (**a**) THz-DTU measurements, showing a good separation of the peaks for the 443-μm and 256-μm papers, but an increasing overlap for paper thicknesses of 164 μm and thinner. (**b**) THz-TOP measurements of the same samples, showing seemingly good separation for all measurements, but a large discrepancy in the estimated thickness for the thinner samples.

**Figure 6 sensors-22-01549-f006:**
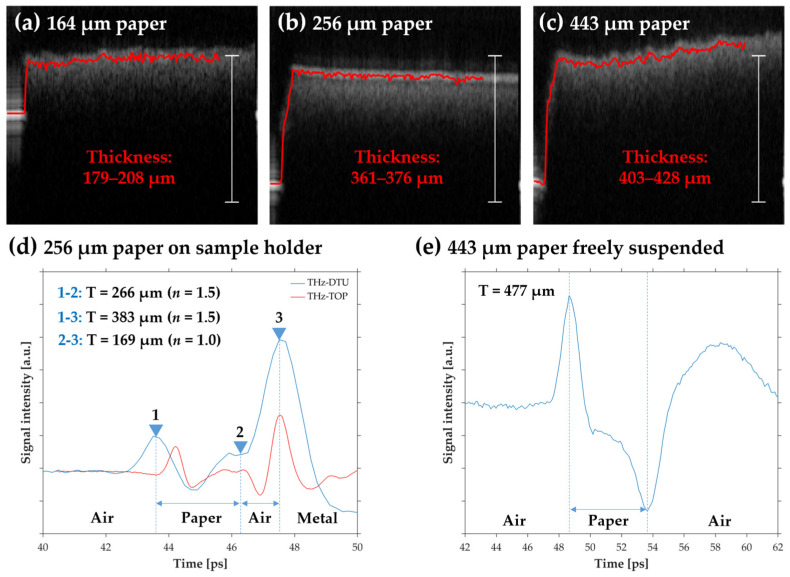
Investigation into the THz overestimation and proof-of-concept demonstration of thickness measurement in freely suspended paper. (**a**–**c**) OCT thickness measurements using the distance from the sample holder to the paper surface. Scale bars represent 0.5-mm thickness in air. (**d**) Analysis of the 256-μm paper THz traces indicating presence of an air gap. (**e**) Proof-of-concept demonstration of THz thickness measurement in freely suspended 443-μm paper sample.

**Figure 7 sensors-22-01549-f007:**
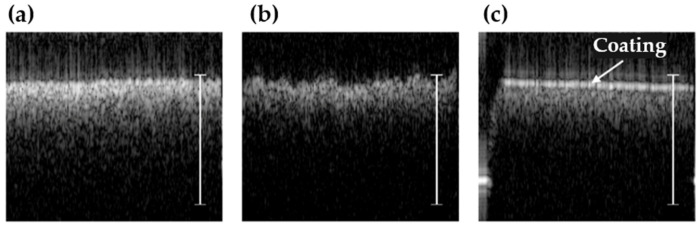
Three paper samples with significantly different surface finishes visualized using OCT. (**a**) Single B-scan of 100-μm printing paper sample. (**b**) Single B-scan of 443-μm packaging paper sample. (**c**) Single B-scan of 256-μm high-gloss paper sample. Scale bars represent 0.5-mm thickness in air.

**Figure 8 sensors-22-01549-f008:**
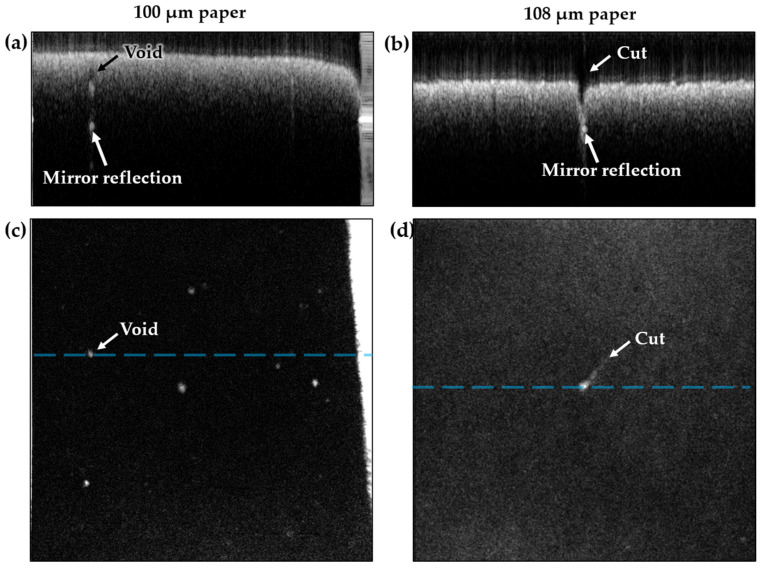
Paper defect detection. (**a**) Average of five B-scans of a 100-μm paper sample with internal voids resulting in a strong reflection from the bottom of the void and from the mirror. (**b**) Average of five B-scans of a cut in a 108-μm paper sample, revealing the mirror below. (**c**,**d**) Top-view volume projections of the subsurface reflections due to voids and the cut, respectively.

**Figure 9 sensors-22-01549-f009:**
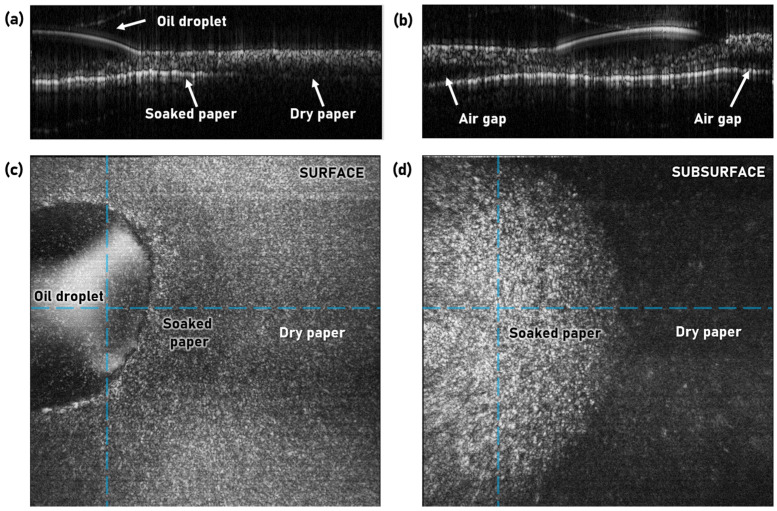
OCT imaging of oil contamination on 68-μm paper sample. (**a**,**b**) Orthogonal B-scans near the oil droplet corresponding to the horizontal and vertical dashed lines seen in (**c**,**d**), respectively. (**c**,**d**) Top-view volume projections of the sample surface and subsurface region (excluding the mirror), respectively.

**Table 1 sensors-22-01549-t001:** Measured paper thicknesses using the digital caliper, OCT, and THz TDS compared to reference values.

Paper Thickness [μm]							
Reference	68	72	83	90	164	256	443
**Digital caliper**	69.3 ± 1.8	72.5 ± 1.1	85.2 ± 1.2	93.8 ± 1.0	164.2 ± 2.7	255.5 ± 5.5	416.0 ± 14.7
**OCT Method 1**	69.6 ^a^ ± 3.2	77.4 ± 2.8	117.3 ± 3.3	105.7 ± 4.3	--	--	--
**OCT Method 2**	52.4 ^a^ ± 5.6	70.5 ± 3.7	80.0 ± 7.4	97.9 ± 6.1	--	--	--
**THz-DTU**	148.7 ± 11.0	--	186.0 ± 5.3	--	298.3 ± 5.0	372.0 ± 3.5	442.3 ± 9.5
**THz-TOP**	88 ± 4 ^b^	--	117 ± 4 ^b^	--	189 ± 4 ^b^	322 ± 4 ^b^	439 ± 4 ^b^

^a^ Average of six different measurements. ^b^ Specified instrument accuracy.

## Data Availability

The data presented in this study are available on request from the corresponding author. The data are not publicly available due to significant storage requirements.

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
