# Peer review of "Non-Contact Paper Thickness and Quality Monitoring Based on Mid-Infrared Optical Coherence Tomography and THz Time Domain Spectroscopy"

_sensors, 2022, doi:10.3390/s22041549_

Round 1

Reviewer 1 Report

In the current manuscript the authors presented a study to investigate paper thickness and defects by using two optical imaging techniques. First an optical coherence tomography (OCT) setup operating in the mid-infrared wavelength region was utilized. The setup provided an axial resolution of eight μm and a transversal resolution of thirty μm. The second technique was THz Time Domain Spectroscopy (THz TDS). Here, two setups, one in house build prototype and a commercial setup were utilized. The prototype had an axial resolution of 150 μm and a lateral resolution of 1 mm and the commercial setup 50 μm and 1 mm, respectively. A set of paper types with thicknesses from 68 - 443 μm were investigated. Further using the OCT setup diverse types of defects and oil contamination were investigated.

In general, the manuscript is well written, however, we have several major and minor comments that would strengthen the article.

  • The whole introduction section just contains a single citation; therefore, most sentences are not supported by any literature. Further it would be interesting for the reader to elaborate in more detail which other techniques have been used and their advantages and limitations.
  • The motivation for using two THz TDS setups is not clear to the reader.
  • The authors stated that: To eliminate air gaps between the paper and mirror, the paper was taped firmly against the mirror. As the authors write in the introduction, the aim of this work is to find a non-contact imaging technique to investigate paper thickness in a production scenario. Is the taping procedure applicable for that?
  • From the white light images in Fig.3, the cellulose composition of the used paper types looks quite different. Did the authors see some influences (better or worse penetration of the light) on the accuracy of the measurements due to these differences?
  • Why is it needed to average over 200 B-scans to obtain the thickness measurements? Did the authors optimize the amount of data used for the thickness evaluation? For a real production scenario an imaging time of 36 seconds seems exceptionally long, can the authors elaborate this point in more details?
  • If the 200 B-scans were not acquired repeatedly in one position, would it be enough to have an average thickness value for quality control?
  • For the THz TDS, ten measurement points were distributed over a range of 5 mm. If the aim is to detect surface thickness changes, would this sparse sampling technique miss some key features (like the last question about quality control)? Again, the long measurement time seems not applicable for a real production scenario.
  • Does the refractive index value of 1.5 in equation in line 171 might vary depending on the composition of the paper?
  • From Table 1 it seems that OCT is only suitable for paper thicknesses below 70 μm and the THz TDS setups for thicknesses above 356 μ Is there a benefit of combining the setups?
  • For the paper quality inspection in thick papers: If the aim is to analyze thickness of the papers with THz TDS and inspect the quality with OCT, wouldn’t it be enough to add a standard OCT setup?
  • The authors stated that: By comparing the results in Table 1, it is evident that the zero-crossings method yielded thickness values that are closer to the reference values. Could this also be an effect of the different axial resolutions provided by the two setups? Further the author states that: In both THz-DTU and THz-TOP, the thicknesses are mostly overestimated, so one attributing factor is the possibility of a small gap of air between the paper and the metal. Wouldn’t the air gap be extremely hard to avoid in a real word scenario?
  • For the discussion, it would be interesting to add a comparison to other techniques and elaborate further steps to improve the accuracy of the measurements and a real-world application of the setups.
  • What is the conclusion of this work?
  • What does the white line in Fig 6 indicate in (a) – (c)?
  • In Fig. 6 – 8 scale bars are missing. This information is crucial to know how deep the light can penetrate the thick paper.
  • It would be interesting to also include the results from the standard single-sided laser triangulation method, as mentioned in the methods part 2.3, to table 1 and discuss the improvements shown with the two techniques.

Author Response

Dear Reviewer,

We would like to thank you for taking the time to review our manuscript. Please find the attached response to all comments and suggestions.

Best regards, on behalf of all authors,

Christian Rosenberg Petersen

Reviewer 2 Report

In this paper, two optical sensing techniques are investigated for non-destructive, non-contact characterization of paper thickness, surface roughness, and production defects. The first technique is optical coherence tomography based on a mid-infrared supercontinuum laser, which can cover thicknesses from ~20-90μm and provide information about the surface finish. And the second technique is terahertz time domain spectroscopy, which can be used in paper samples with a thickness of up to 443μm. Detection of subsurface voids, cuts, and oil contamination was also demonstrated, it is of great significance for on-line monitoring of quality parameters in industrial paper production. It is recommended to publish this manuscript.

Author Response

Dear Reviewer,

We would like to thank the reviewer for the positive feedback regarding the significance of this work, and for the recommendation.

Best regards, on behalf of all authors,

Christian Rosenberg Petersen

Reviewer 3 Report

The paper reports about an interesting application of Mid-infrared Optical Coherence Tomography (OCT) and THz Time domain spectroscopy (TDS). The paper is well organized and well written, except for several typing errors which will probably be fixed by the editorial office. However, before publication, I would like the authors to address my following questions and comments:

  1. Please quantify the stability of the supercontinuum (SC) source used for the OCT studies. Many SC sources exhibit long-term instability problems that would degrade the quality of OCT images.
  2. Please mention explicitly which incoherent light source is used for the Quasi-TDS measurements.
  3. The authors motivate their studies with application in industrial processes. However, I doubt that the procedure under which the paper has to be taped against the mirror before the measurement and the measurement speed are compatible with industrial standards. As this affects the motivation of the paper, I would like the authors to discuss this issue in more detail.    
  4. Please analyze more deeply why THz-DTU and THz-TOP overestimate the thicknesses. Can the assumed small air gap be confirmed by other measurements? What about the uncertainty about the refractive index?

Author Response

(The authors gave the same response as above.)

Round 2

Reviewer 1 Report

We thank the authors for the detailed revision of their manuscript. We believe the additional information provided improved the manuscript. However, there are still some typing errors which could be fixed and additionally we believe addressing the following two open issues would further strengthen the manuscript before publication:

(1) The authors stated that the scale bar in Fig. 7 was omitted. However, the scale bar is still in the image, and we believe as such it would be nice to add a length description.

(2) The authors state in R.1.16: “As such, the two new methods do not provide an improvement in terms of accuracy, but rather new functionality. We highlight this in the discussion.”  We believe the direct impact of this work is still questionable. The motivation of this work was to find an alternative and more accurate solution to single-sided laser triangulation. Both presented techniques and their combination cannot provide this desired improvement. It would be great if the authors could clarify this point even more in the discussion.

Author Response

We thank the reviewer for the constructive feedback. Please see our response and revisions made in the attached letter.
